

# Hemodynamic outcomes in patients undergoing bidirectional cavopulmonary connection with additional or antegrade pulmonary blood flow: a single-centre retrospective study

Kernfan Puthikitakawiwong[1], Chodchanok Vijarnsorn[1], Teerapong Tocharoenchok[2], Prakul Chanthong[1], Kritvikrom Durongpisitkul[1], Paweena Chungsomprasong[1], Supaluck Kanjanauthai[1], Ploy Thammasate[1], Thita Pacharapakornpong[1], Jarupim Soongswang[1], Kriangkrai Tantiwongkosri[2], Ekarat Nitiyarom[2] and Thaworn Subtaweesin[2]

[1] Department of Pediatrics, Faculty of Medicine Siriraj Hospital, Mahidol University, Bangkok, Thailand
[2] Department of Surgery, Faculty of Medicine Siriraj Hospital, Mahidol University, Bangkok, Thailand

Corresponding author
Chodchanok Vijarnsorn,
cvijarnsorn@yahoo.com

## ABSTRACT

**Background.** The bidirectional cavopulmonary connection (BCPC) is a pivotal stage in the surgical palliation of single-ventricle patients. However, there is ongoing debate regarding the benefits and drawbacks of BCPC with additional or antegrade pulmonary blood flow (AAPBF) in optimizing the subsequent stage—total cavopulmonary connection (TCPC).

**Objective.** To determine the influence of BCPC with AAPBF on pulmonary artery growth and hemodynamic outcomes.

**Methods.** A retrospective review was conducted of 167 single-ventricle patients who underwent BCPC at Siriraj Hospital between 2006 and 2022. Patients were categorized into two groups based on AAPBF status: group 1 (with AAPBF, $n = 44$) and group 2 (without AAPBF, $n = 123$). Variables from pre-BCPC and pre-TCPC cardiac catheterization—including pulmonary artery growth, McGoon ratio, Nakata index, mean pulmonary arterial pressure (mPAP), ventricular end-diastolic pressure (EDP), and indexed pulmonary vascular resistance (PVRi)—were compared between groups. Pulmonary artery branch $z$-scores were analyzed and adjusted using analysis of covariance (ANCOVA). Survival rate, all-cause mortality, and incidence of atrioventricular valve regurgitation (AVVR) and pulmonary arteriovenous malformation (AVM) were also assessed.

**Results.** Median ages at pre-BCPC assessment were 1.06 years (group 1) and 2.17 years (group 2), and at pre-TCPC assessment were 6.19 and 7.27 years, respectively. Median age at BCPC operation was similar between groups (1.58 *vs.* 1.51 years). Over a median follow-up of 64.33 months, group 1 showed significantly greater increases in right and left pulmonary artery size compared to group 2 (RPA: 3.27 *vs.* 1.6 mm ($p = 0.019$); LPA: 2.38 *vs.* 0.88 mm ($p = 0.004$)). The Nakata index increased in group 1 but decreased in group 2 (26.70 *vs.* −84.67 mm$^2$/m$^2$, $p < 0.001$). Z-scores confirmed significant growth in both pulmonary arteries when adjusted for body surface area ($p < 0.001$). No significant differences were found in pre-BCPC mPAP (16 *vs.* 15 mmHg, $p = 0.38$), EDP (12 *vs.*
12 mmHg, $p = 0.584$), or PVRi (1.77 *vs.* 2.03 WU m$^2$, $p = 0.890$). Survival rates did not differ significantly between groups ($p = 0.350$).

**Conclusions.** BCPC with AAPBF effectively promotes pulmonary artery growth without adversely affecting ventricular volume loading or pulmonary artery pressure. Further investigation into the development of arteriovenous malformations is recommended.

## INTRODUCTION

In recent decades, a growing number of adults worldwide have survived childhood undergone a Fontan or total cavopulmonary connection (TCPC) procedure (*Reddy et al., 2020*). Cardiac surgical techniques have been evolving to some extent but are still largely unsupported by high-quality evidence of optimal outcomes. Interstage palliative surgeries, such as bidirectional cavopulmonary connection (BCPC), have become a pivotal staged procedure that prepares single ventricle patients for TCPC operation (*Glenn & Patino, 1954*; *Glenn, 1958*; *Konstantinov & Alexi-Meskishvili, 2000*). Candidacy for TCPC operation depends on low pulmonary vascular resistance, good ventricular function, proper pulmonary artery (PA) size, and a well-functioning atrioventricular valve (AVV) (*Venna, Cetta Jr & d'Udekem, 2021*).

The question then arises: how can these conditions for TCPC candidacy be attained? The prospect of a failed TCPC is a nightmare scenario accompanied by multisystem complications. Therefore, BCPC operation is the ultimate long-term palliation for some patients (*Calvaruso et al., 2008*; *Miyaji et al., 1996*). What if BCPC with antegrade or additional pulmonary blood flow (AAPBF) could be used to better prepare patients in terms of promoting PA growth, improving oxygen saturation and preventing pulmonary arteriovenous malformation (AVM) (*Miyaji et al., 1996*; *Demirtürk et al., 2013*; *Ferns et al., 2013*; *Berdat et al., 2005*; *Baek et al., 2021*)? A counterargument to performing BCPC with AAPBF is the potential increase in adverse outcomes, including a higher incidence of pulmonary hypertension, deterioration of ventricular function, and exacerbation of AVV regurgitation (*Ferns et al., 2013*; *Berdat et al., 2005*; *Baek et al., 2021*). One study reported an increased incidence of AVV repair and/or replacement associated with maintaining AAPBF (*Davidson et al., 2023*). The current study hypothesizes that performing BCPC with AAPBF can promote adequate PA growth, thereby optimizing the opportunity for patients to become suitable candidates for TCPC procedure. Therefore, this study aimed to evaluate the effect of AAPBF on PA growth and hemodynamics, with the hope of improving TCPC-related outcomes.

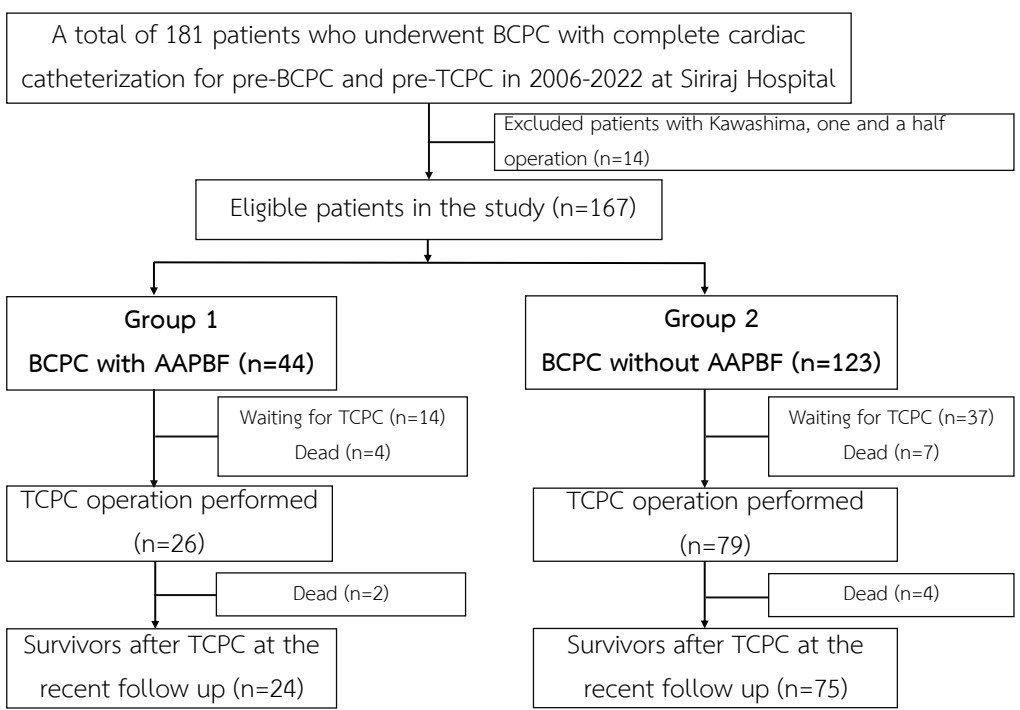

**Figure 1 Flow of the study.** AAPBF, additional or antegrade pulmonary blood flow; BCPC, bidirectional cavopulmonary connection; TCPC, total cavopulmonary connection.

## PATIENTS AND METHODS

### Patient selection

The present study was a single-center, retrospective cohort analysis using a hospital database from a large referral cardiac center in Thailand. The study had been approved by the Siriraj IRB, Faculty of Medicine Siriraj Hospital, Mahidol University (SIRB protocol No. 905/2565 (IRB1), COA no. Si 045/2023) since January 17, 2023. The requirement of informed consent was waived due to the minimal risk posed to the participants. Between 2006 and 2022, a total of 181 patients underwent BCPC and provided complete pre-BCPC and pre-TCPC catheterization data. After excluding patients with one-and-a-half ventricle physiology who underwent a Kawashima operation ($n = 14$), 167 patients were eligible for the analysis (Fig. 1).

### Data collection

Electronic medical records of patients were retrospectively reviewed. The patients' baseline cardiac characteristics, anatomical details, operation details, and follow-up data were explored. Pre-BCPC and pre-TCPC cardiac catheterization data, including diameter of branch PAs (right PA (RPA), left PA (LPA)), ventricular end-diastolic pressure (EDP), and mean pulmonary arterial pressure (mPAP), were collected. The size of the RPA and LPA were measured using an angiogram at the hilum. The Nakata index and McGoon ratios were calculated from the catheterization data. In a supplementary analysis, RPA and LPA
sizes were converted to $z$-scores according to *Lopez et al. (2017)* and analyzed. Information on ventricular function, AVV function, and AVM was also obtained. Oxygen saturation levels were recorded at various stages for intergroup comparisons, and thrombosis and bleeding events were documented. Patients with missing cardiac catheterization data were excluded.

## Surgical techniques

A systemic-to-pulmonary shunt, Norwood, or PA banding procedure was performed based on the patient's cardiac physiology. Bilateral BCPC was conducted in patients with bilateral superior vena cava (SVC). Cardiopulmonary bypass was implemented with or without aortic cross-clamp techniques. The decision to maintain AAPBF was based on a McGoon ratio of <2 during pre-BCPC cardiac catheterization or the presence of bilateral SVC to prevent thrombus formation. During the intraoperative period, AAPBF was adjusted to maintain a Glenn pressure ≤15 mmHg and a systemic oxygen saturation ≥80%. Sources of AAPBF were main PA that was not ligated (native or banded), an open shunt, or an not ligated patent ductus arteriosus (PDA).

## Statistical analysis

Statistical analyses were performed using SPSS Statistics version 22 (IBM Corp, Armonk, NY). Baseline characteristics were reported as frequencies with percentages, medians with interquartile ranges, or means with standard deviations where appropriate. Categorical variables were compared using the Pearson chi-square or Fisher's exact test. The student $t$ test or Mann–Whitney $U$ test was used to conduct between-group comparisons for continuous variables. Patients were divided into two groups based on the preservation of AAPBF. Comparisons were made between the pre-BCPC and pre-TCPC variables. The test of analysis of covariance (ANCOVA) was used to determine changes in RPA and LPA size, McGoon ratio, and Nakata index by adjusting baseline differences prior to analyzing the mean differences between groups. A $p$ value <0.05 was considered statistically significant.

# RESULTS

## Baseline characteristics

The cohort of 167 patients was divided into two categories based on preservation of AAPBF: group 1 underwent BCPC with AAPBF ($n = 44$), and group 2 underwent BCPC without AAPBF ($n = 123$; Fig. 1). Baseline characteristics were not significantly different (Table 1), with a comparable proportion of presenting physiology: pulmonary obstruction (86.4% in group 1, 81.3% in group 2), systemic obstruction (2.3% in group 1, 6.5% in group 2), and excessive pulmonary blood flow (11.4% in group 1, 12.2% in group 2; $p = 0.45$, $p = 0.45$, $p = 0.89$, respectively). The cardiac morphologies are demonstrated in Fig. 2. The initial ventricular function and AVV regurgitation were not statistically different between group 1 and 2.

## BCPC peri-operative outcomes

Peri-BCPC characteristics were mostly similar between groups (Table 2). It is important to highlight that our patients population underwent the BCPC procedure at a relatively

**Table 1  Patients' characteristics (n = 167).**

|  | Group 1 with APBF (n = 44) | Group 2 Without APBF (n = 123) | p-value |
|---|---|---|---|
| Gestational age (<37 weeks) | 5 (11.4%) | 10 (8.1%) | 0.532 |
| Low birth weight (<2,500 g) | 8 (18.2%) | 15 (12.2%) | 0.324 |
| SpO$_2$ (%) | 84 (72–89) | 80 (70–86) | 0.084 |
| Heterotaxy syndrome | 17 (38.6%) | 51 (41.5%) | 0.743 |
| Right isomerism | 11 (25%) | 28 (22.8%) | 0.764 |
| Initial good ventricular function | 44 (100%) | 123 (100%) | – |
| Initial moderate to severe AVVR | 4 (9.1%) | 17 (13.8%) | 0.417 |
| Atrial situs |  |  | 0.532 |
| Normal | 24 (54.5%) | 75 (61%) |  |
| Situs inversus | 6 (13.6%) | 10 (8.1%) |  |
| Common atrium | 14 (31.8%) | 38 (30.9%) |  |
| Confluence pulmonary artery | 44 (100%) | 121 (98.4%) | >0.99 |
| Operation before BCPC |  |  |  |
| Systemic to pulmonary artery shunt | 13 (29%) | 65 (53%) |  |
| Ductal stenting | – | 3 (2%) |  |
| Pulmonary artery banding | 7 (16%) | 14 (11%) |  |
| RV to pulmonary artery shunt | – | 5 (4%) |  |
| Norwood operation | – | 1 (0.8%) |  |
| TAPVR repair | 2 (4%) | 5 (4%) |  |
| Aortic arch repair | – | 2 (1.6%) |  |
| No previous surgery | 21 (47%) | 33 (26%) |  |

Notes.

Data represented as mean ± SD, median (IQR; p25-p75), and n (% within column).

*Statistical significance at p-value <0.05.

AAPBF, , additional or antegrade pulmonary blood flow; AVVR, atrioventricular valve regurgitation; SpO$_2$, oxygen saturation; MAPCAs, major aortopulmonary collateral arteries.

advanced age. Cardiopulmonary bypass time and operative time in group 1 (with AAPBF) were significantly shorter than group 2 (without AAPBF; $p < 0.05$). The source of AAPBF included stenotic main PA ($n = 37$, 84.1%), PDA ($n = 4$, 9.1%), and modified Blalock-Thomas-Taussig shunt (MBTS; $n = 3$, 6.8%). The PDAs were left untouched, with sizes ranging from two to three mm. Of the three patients with MBTS shunts, two had shunts banded with an internal diameter of two mm, while the third had an unmodified MBTS shunts of 2.5 mm.

Three patients (6.8%) in group 1 (with AAPBF) and seven patients (5.7%) in group 2 (without AAPBF) required a reoperation. Neither operative nor in-hospital mortalities were recognized in either group. Two patients (4.5%) in group 1 and four patients in group 2 (3.3%) experienced pleural effusion in the postoperative period immediately following the BCPC procedure, with no statistically significant difference between the groups ($p = 0.654$).

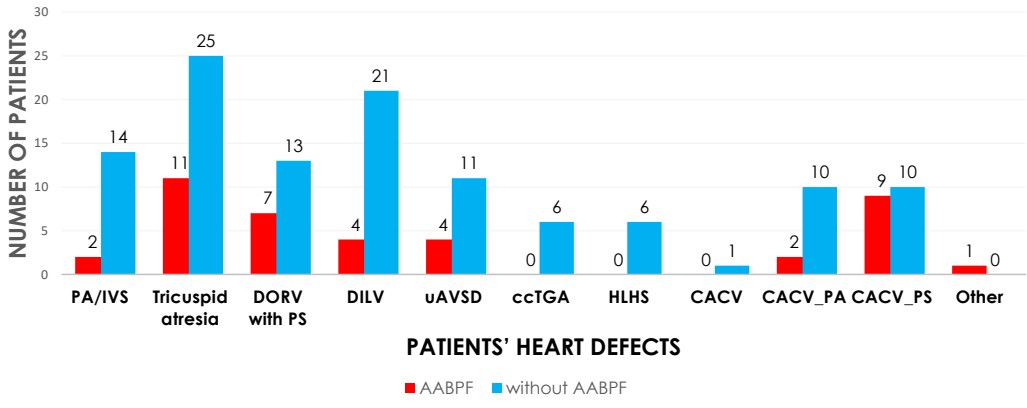

**Figure 2  The cardiac morphologies in operated patients.** PA/IVS, pulmonary atresia with intact ventricular septum; DORV, double outlet right ventricle; PS, pulmonary stenosis; DILV, double inlet left ventricle; uAVSD, unbalanced atrioventricular septal defect; ccTGA, congenital corrected transposition of great arteries; HLHS, hypoplastic left heart syndrome; CACV; common atrium with common ventricle; PA, pulmonary atresia.

**Table 2  Perioperative data and early outcomes of BCPC ($n = 167$).**

| | Group 1 APBF ($n = 44$) | Group 2 Without APBF ($n = 123$) | *p*-value |
|---|---|---|---|
| Age at operation (year) | 1.58 (1.15–2.93) | 1.51 (1.06–3.36) | 0.673 |
| Male sex | 29 (65.9%) | 73 (59.2%) | 0.444 |
| Weight (kg) | 9.95 (7.84–12.5) | 8.79 (7.6–12.3) | 0.209 |
| Bypass time (min) | 69.07 ± 48.49 | 96.02 ± 47.13 | 0.001[*] |
| Operating time (min) | 158.3 ± 61.57 | 193.52 ± 82.75 | 0.011[*] |
| Cross-clamping time (min) | 9.91 ± 20.67 | 8.86 ± 17.84 | 0.803 |
| CVP (mmHg) | 15.48 ± 4.95 | 14.67 ± 5.14 | 0.379 |
| Inotropic support time (h) | 67 (31–99.5) | 49.5 (25.25–103.5) | 0.750 |
| Pulmonary vasodilator usage time (h) | 23.38 ± 52.54 | 15.98 ± 40.8 | 0.367 |
| Duration of ventilatory support (h) | 6.5 (2–22.5) | 7 (4–20) | 0.406 |
| Post-operative SpO$_2$ (%) | | | |
| Immediate | 87.57 ± 10.2 | 87.73 ± 15.7 | 0.951 |
| 24 h postoperative | 84.66 ± 5.55 | 84.3 ± 6.32 | 0.740 |
| Before discharge | 84.09 ± 4.84 | 82.47 ± 5.22 | 0.076 |
| ICU stays (day) | 4 (2–5) | 3 (2–5) | 0.631 |
| Hospital length of stays (day) | 11 (9–17) | 11 (9–15) | 0.881 |

**Notes.**

Data represented as mean ± SD, median (IQR; p25-p75), and n (% within column).

*Statistical significance at *p*-value < 0.05.

AAPBF, additional or antegrade pulmonary blood flow; SpO$_2$, oxygen saturation; BCPC, bidirectional cavopulmonary connection; CVP, central venous pressure; ICU, intensive care unit.

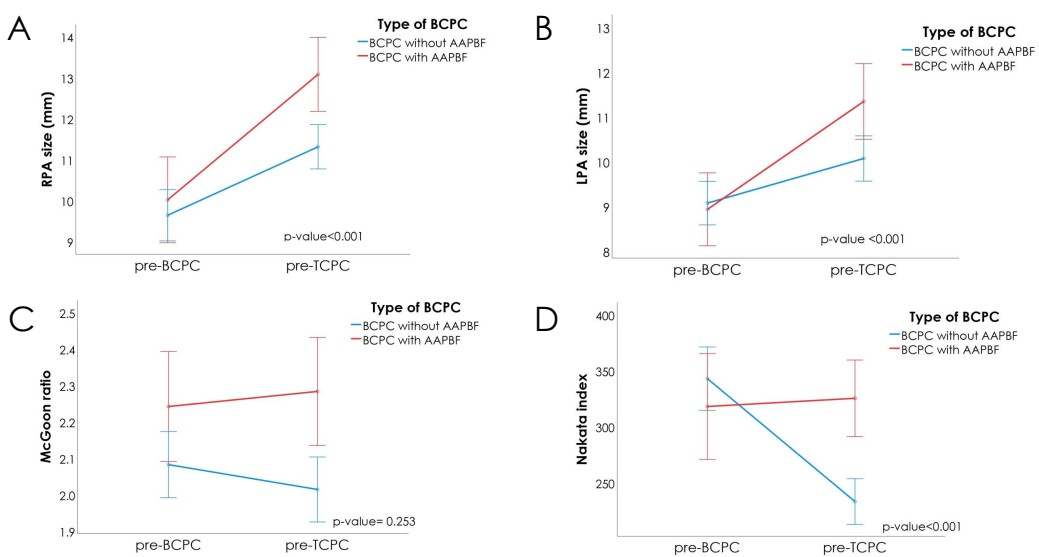

**Figure 3 Changes during the pre-BCPC and pre-TCPC interval.** (A) RPA in both groups increased (more noticeably in group 1). (B) Similarly, the LPA in both groups increased (with a steeper incline in group 1). (C) Accordingly, the McGoon ratio increased in Group 1 while decreasing in Group 2; however, the difference did not reach statistical significance. (D) the Nakata index increased in group 1 and decreased in group 2. AAPBF, additional or antegrade pulmonary blood flow; BCPC, bidirectional cavopulmonary connection; TCPC, total cavopulmonary connection.

## Change in pulmonary size, EDP, and mPAP

Figure 3 and Table 3 present significant changes in absolute pulmonary artery size after applying Bonferroni correction with the significance threshold set at $p = 0.017$. The time intervals between the two assessments were $54.77 \pm 25.63$ months for group 1 and $69.39 \pm 33.60$ months for group 2 ($p = 0.011$), therefore further analysis using ANCOVA was used to adjusted the baseline different. Relative to their original sizes, the RPA increased by 35% in group 1 (with AAPBF) compared to 18% in group 2 (without AAPBF; $p = 0.001$), while the LPA increased by 26% in group 1 *versus* 10% in group 2 ($p = 0.011$). Although the initial $p$-value for RPA change ($p = 0.019$) did not initially meet this Bonferroni-adjusted threshold, further analysis using ANCOVA revealed a statistically significant difference ($p < 0.001$). The $z$-scores for RPA and LPA also showed significant increases in size and rates of changes (Table S1). Furthermore, the z-scores clearly indicated a proportional deterioration in pulmonary artery growth relative to somatic growth in patients without AAPBF. There was a marked decline in the Nakata index in group 2 ($-29\%$ of the original index; $p < 0.05$) compared to a 9.4% increase in group 1 ($p < 0.05$). Although the McGoon ratio was not statistically significant between the groups, follow-up analysis using ANCOVA revealed statistically significant differences across all pulmonary arteries growth parameters, as presented in Table S2.

An insignificant increase in ventricular EDP when compared between groups was noted in group 1 (with AABPF) even though a decrease was observed in group 2 (without AABPF; $p = 0.219$). A decrease in mPAP was observed within groups for both groups, with a more

**Table 3  Changes in pulmonary size, EDP, and mPAP ($n = 167$).**

| | Group 1 With AAPBF ($n = 44$) | | | Group 2 Without AAPBF ($n = 123$) | | | p-value |
|---|---|---|---|---|---|---|---|
| | Pre-BCPC | Pre-TCPC | p-value | Pre-BCPC | Pre-TCPC | p-value | |
| RPA size (mm) | 9.3 (7.31–11.98) | 12.38 (10.35–15.68) | 0.001*** | 8.9 (7.28–11.0) | 11 (9.33–12.90) | <0.001*** | |
| ΔRPA | | 3.27 (0.52–5.07) | | | 1.60 (−0.05–3.53) | | 0.019*** 0.001** |
| RPA growth rate (mm/month) | | 0.06 (0.01–0.11) | | | 0.03 (−0.01–0.51) | | <0.001*** |
| LPA size (mm) | 9.15 (7.07–10.15) | 11.2 (8.75–12.75) | 0.011*** | 8.5 (7.07–10.80) | 9.7 (8.25–11.20) | <0.001*** | |
| ΔLPA | | 2.38 (0.96–4.48) | | | 0.88 (−0.63–2.93) | | 0.004*** 0.002** |
| LPA growth rate (mm/month) | | 0.05 (0.02–0.06) | | | 0.01 (−0.01–0.39) | | <0.001*** |
| McGoon ratio | 2.21 (1.80–2.60) | 2.23 (1.83–2.76) | <0.001*** | 2 (1.75–2.30) | 1.96 (1.75–2.20) | <0.001*** | |
| ΔMcGoon ratio | | 0.035 (−0.37–0.50) | | | −0.04 (−0.40–0.20) | | 0.253 0.009*** |
| Nakata index (mm²/m²) | 283.93 (176.07–421.61) | 317.43 (215.65–400.09) | 0.034*** | 291.83 (229.06–451.56) | 217.85 (170.81–268.64) | <0.001*** | |
| ΔNakata index | | 26.7 (−97.69–123.63) | | | −84.67 (−189.75–14.28) | | <0.001*** <0.001** |
| Nakata index change rate (mm²/m² per month) | | 0.64 (−2.18–2.02) | | | −1.38 (−2.88–0.19) | | <0.001*** |
| mPAP (mmHg) | 18 (15–20.50) | 16 (13–18) | 0.006* | 18 (15–20.75) | 15 (13–17) | 0.182 | |
| ΔmPAP | | −2 (−4.5–1) | | | −3 (−7–1) | | 0.118 |
| Ventricular EDP (mmHg) | 12 (10–13) | 12 (10–14.75) | 0.046* | 12 (10–14) | 12 (10–13) | 0.164 | |
| ΔEDP | | 1 (−2–4) | | | −1 (−3–3) | | 0.219 |
| PVRi (WU m²) | 2.03 (1.28–2.6) | 1.77 (1.03–2.33) | 0.567 | 2.03 (1.21–2.88) | 1.65 (0.99–2.50) | 0.140 | |
| ΔPVRi | | −0.27 (−1.18–0.83) | | | −0.41 (−1.35–0.47) | | 0.190 |

**Notes.**

Data represented as mean ± SD, median (interquartile range; p25-p75), and n (% within column).

*Statistical significance at *p*-value < 0.05.

**Statistical significance at *p*-value < 0.05 by ANCOVA.

***Statistical significance at *p*-value < 0.017 (Bonferroni correction).

****Statistical significance at *p*-value < 0.025 (Bonferroni correction).

AAPBF, additional or antegrade pulmonary blood flow; SpO₂, oxygen saturation; BCPC, bidirectional cavopulmonary connection; TCPC, total cavopulmonary connection; RPA, right pulmonary artery; LPA, left pulmonary artery; PVRi, pulmonary vascular resistance index; EDP, end-diastolic pressure; mPAP, mean pulmonary arterial pressure.

prominent decrease in group 1 ($-11\%$, $p < 0.05$) compared to group 2 ($-17\%$, $p = 0.182$). However, the different did not reach significant when compare between groups ($p = 0.12$). A total of 106 patients in the study (63.5%) had an elevated mPAP of >16 mmHg at the pre-BCPC assessment, with 26 patients from group 1 (59%) and 80 patients from group 2 (65%). Subsequent pre-TCPC data showed that 23 patients in group 1 (52.3%) and 54 patients in group 2 (43.9%) had retained a high mPAP without statistically significant between groups ($p = 0.339$). Of these 23 patients in group 1, 13 patients underwent TCPC with seven fenestration (53.8%). Likewise, 29 patients from group 2 underwent TCPC with 13 fenestration (44.8%). No statistical difference between groups in the proportion of TCPC with fenestration was observed ($p = 0.588$).

Notably, four patients exhibited unfavorable hemodynamic data in pre-TCPC catheterization and were thus identified as high-risk patients not suitable for TCPC. Of these four patients, one patient from group 1 exhibited a high mPAP of >20 mmHg and elevated EDP. The remaining three patients from group 2 experienced pulmonary hypertension and exhibited small sized PA branches.

### Follow-up

Of 167 patients, 105 underwent the complete TCPC procedure. Recent follow-up post-TCPC operation revealed comparable NYHA functional class, AVV regurgitation, and cardiac function in both groups (Table 4). Over a median follow-up duration of 10.24 years (7.81–14.24 years), six patients (13.6%) in group 1 (with AABPF) and 11 patients (8.9%) in group 2 (without AABPF) died due to a variety of causes including septic shock, COVID19 infection, pulmonary hypertension, and massive pulmonary hemorrhage attributed to collateral vessels ($p = 0.420$). At a median follow-up duration of 4.77 years in group 1 and 3.52 years in group 2, protein-losing enteropathy was observed in one patient (3.8%) from group 1 and in two patients (2.5%) from group 2. The incidence of Fontan-associated liver disease was higher in group 1, with three patients (11.5%) affected, compared to two patients (2.5%) in group 2; however, this difference did not reach statistical significance ($p = 0.103$). Similarly, the rate of hospital readmission due to chylothorax did not differ significantly between study groups ($p = 0.621$). The survival rate of patients who underwent BCPC with and without AAPBF is presented in Fig. S1, showing no statistical differences between groups ($p = 0.350$).

## DISCUSSION

The unilateral Glenn operation was first successfully performed in 1958 to alleviate cyanosis and unload the ventricles (*Glenn & Patino, 1954*; *Glenn, 1958*). In 1961, the first BCPC procedure was completed, resulting in better systemic arterial oxygen saturation (*Konstantinov & Alexi-Meskishvili, 2000*). BCPC, however, still results in a much lower pulmonary blood flow (PBF) compared to baseline or post-TCPC, and this decreased PBF jeopardizes PA growth (*Uemura et al., 1995*; *Mendelsohn et al., 1994*; *Slavik, Franklin & Radley-Smith, 1999*; *Kansy et al., 2013*). Therefore, AAPBF was introduced to help promote PA growth (*Ferns et al., 2013*; *Berdat et al., 2005*; *Baek et al., 2021*; *Mendelsohn et al., 1994*; *Van de Wal et al., 1999*).
**Table 4   Long term outcomes following BCPC ($n = 167$).**

| | Group 1 APBF ($n = 44$) | Group 2 Without APBF ($n = 123$) | *p*-value |
|---|---|---|---|
| Archives of TCPC operation | 26 (59.1%) | 79 (64.2%) | 0.545 |
| Await for TCPC operation | 14 (31.8%) | 37 (30%) | 0.830 |
| TCPC operation with fenestration | 8 (18.2%) | 35 (28.5%) | 0.223 |
| Age at TCPC operation (year) | 7.84 ± 2.04 | 8.58 ± 2.25 | 0.107 |
| SpO$_2$ pre-TCPC operation (%) | 81.04 ± 7.42 | 78.90 ± 6.23 | 0.151 |
| Functional class III–IV | 0 (0%) | 4 (3.3%) | 0.574 |
| SpO$_2$ at recent follow-up (%) | 88.84 ± 8.02 | 87.61 ± 9.44 | 0.552 |
| Moderate to severe AVVR | 6 (13.6%) | 20 (16.3%) | 0.638 |
| Significantly reduced EF | 9 (20.5%) | 24 (19.5%) | 0.893 |
| Arrhythmia | 8 (18.2%) | 25 (20.3%) | 0.759 |
| Thrombosis | 2 (4.5%) | 14 (11.4%) | 0.242 |
| Arteriovenous malformation | 1 (2.3%) | 10 (8.1%) | 0.291 |
| Venovenous collateral | 21 (47.7%) | 47 (38.2%) | 0.270 |
| Survival | | | 0.351 |
| 5-year survival (%) | 90.6% | 96.6% | |
| 10-year survival (%) | 83.7% | 93.1% | |
| 15-year survival (%) | 83.7% | 84.9% | |

**Notes.**
Data represented as mean ± SD, and n (% within column).
*Statistical significance at *p*-value < 0.05.
AAPBF, additional or antegrade pulmonary blood flow; AVVR, atrioventricular valve regurgitation; SpO$_2$, oxygen saturation; MAPCAs, major aortopulmonary collateral arteries; BCPC, bidirectional cavopulmonary connection; TCPC, total cavopulmonary connection.

The advantages of AAPBF, including enhanced PA growth and improved oxygen saturation, have been previously reported (*Ferns et al., 2013*; *Baek et al., 2021*; *Kowatari et al., 2020*; *Sughimoto et al., 2015*; *Ferns et al., 2021*). The current study corroborates these findings, revealing a trend toward slightly higher oxygen saturation prior to discharge from the BCPC operation, prior to the TCPC operation, and at the subsequent follow-up when compared to patients without AAPBF. However, the differences observed in this investigation were not statistically significant, contrary to what has been reported in previous studies (*Berdat et al., 2005*; *Chen et al., 2015*). Patients with AAPBF in the current study had slightly larger PA branches before undergoing the BCPC operation, indicating that the underlying reasons for AAPBF were beyond solely the augmentation of PAs. At the pre-BCPC assessment, there were no statistically significant differences in RPA and LPA sizes between the two groups ($p = 0.412$ and $p = 0.971$, respectively). These variables were therefore adjusted for in the subsequent analysis since the interval between two catheterizations were difference. In this study, the modified Blalock-Thomas-Taussig shunt constituted a minority of the cases with AAPBF. This may be attributed to the limitations of the surgical technique, which could potentially increase the risk of PA distortion due to somatic outgrowth relative to the fixed shunt length (*Godart et al., 1998*).

The cardiopulmonary bypass time and total operative time were significantly lower in group 1, likely due to the omission of time-consuming procedures required for pulmonary blood flow elimination ($p < 0.05$ for both comparisons). In the study cohort, there were no significant differences in ICU or hospital length of stay among patients with AAPBF when compare to patients without AAPBF. Similarly, there were no significant differences in the incidence of reoperations and reinterventions during the admission period. These findings were consistent with those of previous studies (*Chen et al., 2015*; *Gray et al., 2007*).

During the interstage period, the cardiac catheterization data showed a greater change in size and rate of growth of both PAs in group 1 (with AAPBF; $p < 0.05$ for all). The analysis of the $z$-scores of RPA and LPA in group 1 also showed significant increases in size and rates of change based on the absolute values and mean differences (Table S1). The findings clearly highlight a significant deterioration in PA growth in group 2 (without AAPBF) relative to overall somatic growth. The Nakata index demonstrated a marked increase in group 1, whereas a significant decline was observed in group 2 ($p < 0.05$ for both). This statistically significant difference persisted across all assessed parameters including RPA and LPA size, the McGoon ratio, and the Nakata index after adjusting for baseline values (Table 3, Table S2, and Fig. 3). Importantly, this statistical significance remained robust after the interval between the pre-BCPC and pre-TCPC assessment, pre-BCPC parameters, age at assessment, and body weight were accounted for using ANCOVA. These findings further underscore the substantial PA growth observed in group 1. Collectively, these results illustrate a preserved capacity for PA growth in proportion to body surface area throughout the evaluation period. This observation is particularly relevant since both parameters would generally be expected to decline over time in univentricular patients (*Kansy et al., 2013*).

Notably, the growth of the RPA size was greater than that of the LPA, even among patients who did not undergo BCPC with AAPBF. These findings align with those of previous studies. For example, *Slavik et al. (1995)* observed a trend towards a reduced postoperative left pulmonary artery size in patients who underwent BCPC without AAPBF. Additionally, this study assessed the association between LPA regression and several operation variables including the type of BCPC (bilateral *vs.* unilateral), pulmonary artery plasty during BCPC, and left pulmonary plasty during BCPC. However, none of these factors showed a statistically significant association with LPA regression. Additionally, a case series by *Javadi et al. (2022)* demonstrated that mean blood flow through the RPA was higher than that through the LPA, as assessed by cardiac magnetic resonance imaging (MRI). These observations led to the hypothesis that factors beyond anatomical structure, particularly those related to fluid dynamics and energy loss associated with various surgical approaches, may contribute to differential pulmonary artery growth.

There is debate as to whether AAPBF poses a risk for pulmonary hypertension (*McElhinney, Marianeschi & Reddy, 1998*; *Chen et al., 2020*). Pulmonary vascular resistance index (PVRi) was increased in group 1 and decreased in group 2 ($p = 0.190$). Moreover, the data from the current study showed a significant reduction in mPAP within group 1 ($p < 0.05$), in accordance with previous reports (*Berdat et al., 2005*; *Caspi et al., 2003*). However, the decline in mPAP was not statistically significant between the groups

($p = 0.118$), suggesting an inconsequential impact on ventricles. Therefore, the current study did not reach a conclusion regarding whether AAPBF significantly affects mPAP. Currently, there is no universal guideline for defining elevated mean pulmonary artery pressure in single ventricle patients. Prior research indicated that an mPAP $\geq 16$ mmHg was associated with adverse events following TCPC (*Kido et al., 2021*). Using this cutoff point, patients with an mPAP $\geq 16$ mmHg following pre-BCPC catheterization decreased after BCPC in both groups. Analysis of the incidence of fenestrated TCPC revealed no significantly higher proportion in group 1 compared to group 2 ($p = 0.223$). This finding may suggest that patients who underwent BCPC with AAPBF potentially exhibited a physiologic status prior to TCPC that is not detrimental when compare to BCPC without AAPBF.

Concerns have been raised regarding the potential impact of AAPBF on ventricular volume loading (*Miyaji et al., 1996*; *Uemura et al., 1995*). Although the current study revealed a significant increase in ventricular EDP between two cardiac catheterizations in group 1 ($p < 0.05$), the change was not statistically significant when the intergroup comparison to group 2 ($p = 0.219$). Further analysis revealed that a LVEDP $\geq 13$ mmHg was significantly associated with mortality in group 2 patients (without AAPBF; $p = 0.026$). In contrast, no significant association was observed between elevated LVEDP and mortality in group 1 (with AAPBF). These findings suggest the presence of additional volume loading in patients with AAPBF; however, this volume did not appear to negatively impact clinical outcomes. These findings are also consistent with previous studies (*Berdat et al., 2005*; *Kowatari et al., 2020*). A higher incidence of AVV failure necessitating surgical intervention in patients with AAPBF has been reported (*Davidson et al., 2023*). At the initial assessment for the current study, moderate to severe atrioventricular valve regurgitation (AVVR) was observed in four patients (9.1%) with AAPBF and 17 patients (13.8%) without AAPBF ($p = 0.417$). A subgroup analysis was performed on patients who exhibited moderate to severe AVVR at both baseline and at the most recent follow-up. There were no statistically significant differences found between the groups regarding AV valve morphology, ventricular morphology, heterotaxy status, or history of prior surgical interventions (Supplementary S3). Interestingly, nine patients (6.3%) without AAPBF required AVV repair following the BCPC operation, whereas none of the patients in the AAPBF group underwent such intervention. Other previously published studies have reported findings consistent with those from the current study (*Ferns et al., 2013*; *Baek et al., 2021*).

Recent research has highlighted the potential benefits of AAPBF, demonstrating its role in mitigating the development of pulmonary AVM and collateral vessel (*Gray et al., 2007*; *Henaine et al., 2013*; *Yoshida et al., 2005*). Abnormal vascular connections between arteries and veins could precipitate systemic desaturation and eventual ventricular dysfunction (*Heinemann et al., 2001*; *Mohammad Nijres et al., 2020*; *Ichikawa et al., 1995*). However, the current study was unable to demonstrate prevention of the development of AVM in AAPBF. Based on our findings, maintaining AAPBF might be beneficial in single ventricle patients with borderline hypoplastic PA to optimize PA growth without jeopardizing the candidacy of TCPC.

## Study limitations

This study is limited by its retrospective design. The evolving perioperative evaluation criteria for both BCPC and TCPC procedures over time rendered some patients ineligible for inclusion. Additionally, certain postoperative complications, such as the duration of chest tube drainage, could not be assessed due to incomplete medical records. The follow-up protocol primarily relied on echocardiography, which may have underestimated pulmonary artery growth at the most recent follow-up assessment. Furthermore, due to the limited sample size, a definitive conclusion could not be drawn on whether AAPBF leads to a better TCPC physiology. To address this limitation, the research team is currently conducting post-TCPC hemodynamic assessments using cardiac MRI.

## CONCLUSIONS

BCPC with AAPBF effectively maintains the indexed pulmonary artery growth without a significant impact on ventricular volume loading, mean pulmonary arterial pressure, or the severity of atrioventricular valve regurgitation. This approach may offer potential benefits for patients who are initially considered suboptimal candidates for TCPC completion, or, alternatively, may support long-term palliation by optimizing BCPC physiology in those who are unsuitable for TCPC procedure.

## ACKNOWLEDGEMENTS

The authors wish to thank the faculty and staff of Cardiovascular Thoracic Surgery, Faculty of Medicine, Siriraj Hospital, for their support and involvement with congenital heart disease patient care. We also acknowledge Dr. Julaporn Pooliam, Clinical Epidemiology Unit, Office of Research and Development, Faculty of Medicine, Siriraj Hospital, and Dr. Pongkhun Panthaneeya MD for their assistance with the statistical analysis. We would like to extend our thanks to Glen Wheeler for his effective proofreading and editing. ChatGPT was used for minor grammar refinements, with all substantive content and analysis conducted by the authors. Final editing was performed by PeerJ language copyediting service.

### Funding

The authors received no funding for this work.

### Competing Interests

The authors declare there are no competing interests.

### Author Contributions

- Kernfan Puthikitakawiwong conceived and designed the experiments, performed the experiments, analyzed the data, prepared figures and/or tables, authored or reviewed drafts of the article, and approved the final draft.

- Chodchanok Vijarnsorn conceived and designed the experiments, performed the experiments, analyzed the data, prepared figures and/or tables, authored or reviewed drafts of the article, and approved the final draft.
- Teerapong Tocharoenchok performed the experiments, analyzed the data, prepared figures and/or tables, authored or reviewed drafts of the article, and approved the final draft.
- Prakul Chanthong performed the experiments, authored or reviewed drafts of the article, provide care for study patients, and approved the final draft.
- Kritvikrom Durongpisitkul performed the experiments, authored or reviewed drafts of the article, provide care for study patients, and approved the final draft.
- Paweena Chungsomprasong performed the experiments, authored or reviewed drafts of the article, provide care for study patients, and approved the final draft.
- Supaluck Kanjanauthai performed the experiments, authored or reviewed drafts of the article, provide care for study patients, and approved the final draft.
- Ploy Thammasate performed the experiments, authored or reviewed drafts of the article, provide care for study patients, and approved the final draft.
- Thita Pacharapakornpong performed the experiments, authored or reviewed drafts of the article, provide care for study patients, and approved the final draft.
- Jarupim Soongswang performed the experiments, authored or reviewed drafts of the article, provide care for study patients, and approved the final draft.
- Kriangkrai Tantiwongkosri performed the experiments, authored or reviewed drafts of the article, provide care for study patients, and approved the final draft.
- Ekarat Nitiyarom performed the experiments, authored or reviewed drafts of the article, provide care for study patients, and approved the final draft.
- Thaworn Subtaweesin performed the experiments, authored or reviewed drafts of the article, provide care for study patients, and approved the final draft.

## Human Ethics

The following information was supplied relating to ethical approvals (i.e., approving body and any reference numbers):

This research was approved by the IRB of Siriraj Hospital, Mahidol University (905/2565 (IRB1) COA no. Si 045/2023).

## Clinical Trial Ethics

The following information was supplied relating to ethical approvals (i.e., approving body and any reference numbers):

This research was approved by the Thai Clinical Trials Registry.

## Data Availability

The raw data is available in the Supplemental Files.

## Clinical Trial Registration

The following information was supplied regarding Clinical Trial registration:

TCTR20230410005.

## Supplemental Information

Supplemental information for this article can be found online at http://dx.doi.org/10.7717/peerj.20021#supplemental-information.

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
