# Peer review of "Hemodynamic outcomes in patients undergoing bidirectional cavopulmonary connection with additional or antegrade pulmonary blood flow: a single-centre retrospective study"

_PeerJ, doi:10.7717/peerj.20021_

## Round 0.1 · original submission · Major Revisions

Reviewer 1 has provided particularly detailed comments.

**Language Note:** The review process has identified that the English language must be improved. PeerJ can provide language editing services - please contact us at [email protected] for pricing (be sure to provide your manuscript number and title). Alternatively, you should make your own arrangements to improve the language quality and provide details in your response letter. – PeerJ Staff

Reviewer 1 ·

Basic reporting

The manuscript presents a well-structured and comprehensive study on the effects of additional or antegrade pulmonary blood flow (AAPBF) in patients undergoing bidirectional cavopulmonary connection (BCPC). However, several areas require refinement to enhance clarity, consistency, and scientific rigor. Below please find some suggestions to address before the manuscript will be considered for publication.

Abstract
No details are provided on how baseline differences were adjusted. Were adjustments made using multivariate analysis, propensity scoring, or another method? It is good for readers to know from abstract what methodology was used, since many readers review abstracts only while searching the databases.
When describing groups, saying "group 1) with and group 2) without" is vague it would be much clearer to label them explicitly as "with AAPBF" and "without AAPBF."
The phrase “retrospectively” is incorrectly used—it should be “respectively” when referring to RPA and LPA growth.

The work encounters numerous grammatical and structural errors, examples of which are given below:
The phrase "Much debate exists about the effectiveness and outweigh of BCPC with additional or antegrade pulmonary blood flow (AAPBF) to their next stage palliation" is grammatically incorrect and unclear. "Outweigh" seems to be misused—do you mean advantages or disadvantages? The transition to TCPC should be framed more clearly.
"Interstage palliative surgeries, such as bidirectional cavopulmonary connection (BCPC), have become a pivotal staged procedures which prepare patients with single ventricle for the designated operation."
o “pivotal staged procedures” → Singular/plural inconsistency (should be “a pivotal staged procedure”).
o “designated operation” → Vague phrasing (should specify TCPC explicitly).
The phrase "Counterargument regarding potential of increase incident of pulmonary hypertension, worsening ventricular function and severity of AVV regurgitation" is grammatically incorrect and lacks detail.

Lines 194-195:
"The advantages of the AAPBF, including enhanced PA growth, improved oxygen saturation, and reduced mortality rates, have been previously reported."
AAPBF itself does not "reduce mortality"; improved PA growth and better hemodynamics may contribute to lower mortality.
Lines 201-203:
"The modified Blalock-Thomas-Taussig shunt represents a minority of the AAPBF possibly due to the limitation in the surgical technique and the potential risk of PA distortion from somatic outgrowth of the shunt length."
This sentence is too long and unclear.
"No different in ICU or hospital stays among patients with AAPBF." (Incorrect grammar)

The phrase “selection bias is hardly avoidable” is grammatically incorrect and vague.
The phrase "We also thanks Glen Wheeler for their effective proofreading and editing.” Is grammatically incorrect.
The sentence "The AI (ChatGPT) was used to correct grammar only." does not follow standard authorship guidelines. If an explanation of authorship is necessary here for transparency, rephrase it:
"ChatGPT was utilized for minor grammar refinements, with all substantive content and analysis conducted by the authors."

In many places the sentences are long (for instance description of patient selection and data analysis) and difficult to follow. Consider breaking it into shorter sentences to improve readability.

Experimental design

The introduction mentions PA growth and oxygenation but does not emphasize long-term outcomes like AVVR, arrhythmias, or thrombosis. These are important findings in the study but seem understated in the introduction. Their mention as the secondary outcomes will place introduction in line with the later discussion.
Additionally, the introduction does not emphasize why pulmonary artery growth is particularly crucial for total cavopulmonary connection (TCPC) candidacy and how AAPBF might influence this.
The introduction presents the positive impact of ABF in a very selective way and does not quote studies that indicate the negative impact of using this strategy. It would be beneficial to cite existing conflicting evidence (e.g., studies that argue against AAPBF).
The sentence "we hypothesize that BCPC with AAPBF can offer a solution for these challenges in selected patients" is vague. Does the study aim to prove that AAPBF improves TCPC eligibility, or just that it does not cause harm?
Patients and methods

While patient selection is well-defined, there is limited discussion on potential selection bias. It is known from experimental animal studies and clinical trials that individuals who have low pulmonary resistance benefit the most from an additional source of blood flow to the lungs. In the study group, ABF appears to have been used more frequently in patients with potentially higher resistance (less developed pulmonary arteries). It remains unclear whether patients with smaller initial pulmonary arteries or better ventricular function were more likely to receive AAPBF, which could confound the results.
Furthermore, although the statistical approach is generally appropriate, there is no mention of whether multiple comparisons correction (e.g., Bonferroni or Benjamini-Hochberg) was applied to mitigate the risk of Type I errors given the numerous outcome comparisons. Additionally, effect sizes (such as Cohen’s d for continuous variables or odds ratios for categorical comparisons) are not reported, making it difficult to assess the clinical relevance of significant differences.

Validity of the findings

Results
The results section largely corresponds with the data presented in tables and figures, but some inconsistencies and gaps need attention. The McGoon ratio, for example, is described as increasing in Group 1 (AAPBF) and decreasing in Group 2, yet the reported p-value (0.253) indicates that this difference is not statistically significant. The text should clarify that while there was a trend toward improved McGoon ratio with AAPBF, it did not reach significance. Similarly, the reduction in mean pulmonary arterial pressure (mPAP) in both groups is discussed, but the lack of statistical significance (p=0.118) is not sufficiently explained. The findings should explicitly state whether this suggests a lack of meaningful impact or if the study was underpowered to detect a difference. Additionally, the proportion of patients undergoing fenestrated TCPC is reported in Table 4 but is not explored in depth in the discussion. Since fenestration can indicate suboptimal Fontan physiology, it would be valuable to assess whether lower fenestration rates in Group 1 suggest a clinical advantage of AAPBF.

Most statistical comparisons only report p-values without effect sizes (e.g., mean differences, confidence intervals).

The follow-up duration between pre-BCPC and pre-TCPC is significantly different between groups (54.77 vs. 69.39 months). Was this difference adjusted for in the analysis? If not, it could be a confounding factor affecting PA growth rates.
The definition of “high mPAP” (>16 mmHg) is somewhat arbitrary—is this based on a standard guideline, or was it chosen for this study?
Were there any early (effusions) or late complications such as protein-losing enteropathy, arrhythmias, or plastic bronchitis? These are critical outcomes for post-TCPC patients but are not mentioned in the study.
Was there a difference in functional status between groups (e.g., NYHA classification, exercise capacity, or quality of life)? The sentence “comparable functional class, AVV regurgitation, and cardiac function” is vague and should be expanded.
The supplementary tables strongly support the main findings but require better integration into the discussion to fully explain the significance of indexed PA growth, McGoon ratio trends, and proportional pulmonary artery development.
The results state that the McGoon ratio "trended upward" in Group 1 but was not statistically significant in unadjusted analysis (p=0.253 in Table 3).
However, after ANCOVA adjustment (Table S2), the difference does reach statistical significance (p=0.009).
The discussion does not mention this important finding, leaving an inconsistency between the unadjusted and adjusted results. The adjusted analysis confirms that AAPBF was associated with significantly better PA-to-aortic size ratios by the time of TCPC.

Table S1 shows that RPA/LPA Z-scores in Group 1 were relatively stable, whereas Group 2 experienced significant declines.
The rate of Z-score change per month was significantly different (p=0.005 for RPA, p=0.003 for LPA).
The discussion does not emphasize that Group 2’s pulmonary arteries were proportionally shrinking relative to body growth.

The results show that absolute size, adjusted means, and growth rates suggest that RPA responded better to AAPBF than LPA. In the non-AAPBF group, LPA regressed more than RPA. This suggests that LPA may be more dependent on additional blood flow for normal growth. Can you explain it based on your material? Does LPA regression correlate with worse Fontan circulation? Could this explain why some TCPC patients require fenestration or develop venovenous collaterals? Should AAPBF be selectively maintained if the LPA is borderline or hypoplastic? Could during (hybrid approach) or post-BCPC interventions (e.g., balloon angioplasty) help balance RPA vs. LPA growth?
The analysis confirms that AAPBF benefits both pulmonary arteries but has a stronger impact on RPA growth. In contrast, the LPA is more vulnerable to regression, particularly in the absence of AAPBF but no statistical comparison between LPA and RPA is provided. Maybe it would be interesting to look if there is a significant difference? Possibly it makes a difference?.

Discussion
The study reports that AAPBF patients had fewer arteriovenous malformations (AVMs) (2.3% vs. 8.1%), but the discussion does not elaborate on whether this supports the hypothesis that AAPBF may mitigate AVM formation. Since AVMs are a key concern in single-ventricle physiology, this finding should be given more attention. Likewise, venovenous collaterals were more frequent in Group 1 (47.7% vs. 38.2%), but no interpretation is provided regarding their impact on long-term hemodynamics. Are these collaterals an adaptive response to altered flow patterns, or do they indicate potential complications such as increased systemic venous congestion? The potential clinical implications should be addressed.
Moreover, the discussion of ventricular volume loading and AVVR could be expanded. The results indicate that ventricular end-diastolic pressure (EDP) was slightly higher in Group 1 before TCPC (p<0.05), but the between-group comparison was not statistically significant (p=0.219). This should be discussed in the context of whether AAPBF introduces additional volume load, particularly since previous studies have raised concerns about its impact on ventricular function. Interestingly, while prior research has suggested that AAPBF may contribute to AVVR progression, the study found that AVV repair was more frequent in Group 2. This contradicts prior concerns, yet the discussion does not explore why. Could differences in baseline valve function or ventricular morphology explain this finding?
The results state that TCPC with fenestration was performed in 18.2% (Group 1) vs. 28.5% (Group 2) (p=0.223). Does this suggest that AAPBF leads to better Fontan physiology, reducing the need for fenestration?

While the absolute PA growth findings are well discussed, the Z-score changes are barely mentioned in the results or discussion. The data from supplementary Table S1 could strengthen the discussion whether Z-score changes confirm the advantage of AAPBF in preserving proportional PA growth.

Additional comments

Study limitations
Were patients with smaller pulmonary arteries or poorer ventricular function more likely to receive AAPBF? If so, this could confound the results. Additionally, perioperative evaluation methods changed over the study period, but there is no mention of what aspects evolved and whether adjustments were made to account for these changes. The reliance on echocardiography for follow-up assessments is also noted as a limitation, but the discussion does not clarify whether interobserver variability was assessed or whether alternative imaging modalities (e.g., MRI or cardiac catheterization) were available for cross-validation.

Conclusion
The statement "could be beneficial for a selected subgroup of these high-risk patients” is vague. Which patients may benefit most: Those with borderline PA size? Those at risk of developing AVMs or poor TCPC candidacy?

Reviewer 2 ·

Basic reporting

no comment

Experimental design

no comment

Validity of the findings

no comment

Additional comments

In their article titled “Hemodynamic results of patients undergoing bidirectional Glenn operation with supplemental or antegrade pulmonary blood flow”, the authors emphasized an important point. Although there are studies on this subject in the literature, I think that some of their significant findings may contribute to the literature.

In general, it is a well planned and discussed article.

I think this is especially important considering that the age of operation is more advanced compared to European and American cases.

It would be useful to share the pre BCPC valvular insufficiency status of the patients in order to compare them with the negative reports in the literature on this subject.

Is there any explanation for the worse survival rates in group 1 in the first years?

Is there any explanation for the longer duration of operation in group 2? If there is, please mention it in the discussion.

It would be useful to revise the article in terms of some grammatical and spelling mistakes.

---

## Round 0.2 · accepted · Accept

We acknowledge that the revisions you submitted in response to the reviewers' comments were appropriate and thoroughly addressed all concerns raised during the peer review process. Based on the scientific merit of your study and the quality of your responses, we are confident that your manuscript meets the standards of PeerJ and is suitable for publication.

Reviewer 1 ·

Basic reporting

The manuscript demonstrates a coherent and professionally structured presentation that conforms to the standard sections expected in scientific reporting. The introduction provides sufficient background to contextualize the study within the broader field of single-ventricle palliation, with appropriate referencing of foundational and recent literature, including both supportive and critical views on the role of additional or antegrade pulmonary blood flow (AAPBF).

Figures and tables are relevant, well-organized, and enhance comprehension of the results. Supplementary materials are appropriately integrated into the main text, and the addition of long-term outcome data further strengthens the completeness of the manuscript. The raw data appear sufficiently detailed to support transparency and reproducibility.

A final professional language review is recommended to ensure full compliance with standards of unambiguous and technically precise English.

Overall, the article is self-contained, presents results directly relevant to the stated hypotheses, and does not appear to be inappropriately subdivided from a larger dataset.

Experimental design

The manuscript addresses a clinically significant and timely research question concerning the role of additional or antegrade pulmonary blood flow (AAPBF) in pulmonary artery development and hemodynamic outcomes following bidirectional cavopulmonary connection (BCPC). The research question is well defined, and the study clearly identifies a knowledge gap related to the long-standing debate on whether AAPBF confers benefits or introduces hemodynamic risk. The authors make a compelling case for the relevance of their investigation within the scope of single-ventricle palliation strategies.

The study is conducted to a high technical standard, using retrospective data from a large, single-center cohort with appropriate ethical approval. Importantly, the revised manuscript demonstrates improved statistical rigor, including the application of Bonferroni correction and ANCOVA for baseline-adjusted comparisons.

The methods section is generally comprehensive, describing patient selection, surgical approaches, catheterization metrics, and follow-up parameters in sufficient detail to allow reproducibility. However, minor clarifications remain warranted. Specifically, while ANCOVA is now correctly applied, the manuscript would benefit from a clearer explanation of how confounders such as time intervals between assessments and baseline PA sizes were normalized or adjusted. Additionally, though trends in RPA vs. LPA growth are discussed, no direct statistical test comparing these two is performed; acknowledging this limitation in the methods or discussion would strengthen transparency.

Overall, the experimental design is robust, appropriate for the clinical context, and executed in a way that supports valid and meaningful interpretation of the data.

Validity of the findings

The findings presented in the manuscript are based on a robust dataset and are analyzed using appropriate statistical methods. The inclusion of a large single-center cohort, detailed longitudinal follow-up, and multiple outcome measures lends credibility to the results. The application of ANCOVA to adjust for baseline imbalances, as well as the use of Bonferroni correction to address multiple comparisons, strengthens the statistical validity. Additionally, the authors have improved the manuscript by integrating supplementary Z-score data, providing a more nuanced view of indexed pulmonary artery growth.

All relevant data appear to be provided, and the results are presented in a transparent and accessible manner. However, while the trends in differential growth between the right and left pulmonary arteries are well described, the absence of a direct statistical comparison limits the strength of this conclusion. Acknowledging this limitation is important.

The conclusions are generally well aligned with the study’s findings and appropriately cautious. The authors correctly avoid overstating non-significant results (e.g., differences in mPAP and fenestration rates). Claims regarding the benefits of AAPBF are supported by the observed improvements in pulmonary artery growth and stable hemodynamics, without extending to unsupported causal inferences. The authors appropriately frame their interpretation within the observational nature of the study.

Overall, the study adds value to the literature by addressing a clinically relevant and previously unsettled question using a rigorous analytic framework. The findings are valid within the constraints of a retrospective design and are presented with a commendable level of transparency and methodological care.

Additional comments

The authors have made substantial and thoughtful revisions that meaningfully improve the manuscript’s clarity, scientific rigor, and clinical relevance. The integration of supplementary data and more sophisticated statistical handling (e.g., ANCOVA, Bonferroni correction) demonstrates responsiveness to prior critique and enhances the robustness of the conclusions.

The addition of follow-up data on long-term complications—such as protein-losing enteropathy, Fontan-associated liver disease, and chylothorax—adds important context.

One notable strength of the revision is the balanced and appropriately cautious interpretation of non-significant results, particularly regarding mPAP, PVRi, and fenestration outcomes. The authors’ transparency in acknowledging statistical limitations—such as the absence of direct inter-artery comparisons—is appreciated.

A final professional language and grammar review is recommended to correct remaining minor errors and improve overall readability.

The study is a valuable contribution to the field of congenital heart surgery and single-ventricle palliation. It offers data-driven insight into a long-standing clinical question and supports future investigation into patient selection for AAPBF in BCPC procedures.